# Modeling Extremes with $d$-max-decreasing Neural Networks

**Ali Hasan**[1]    **Khalil Elkhalil**[2]    **Yuting Ng**[2]    **João M. Pereira**[3]    **Sina Farsiu**[1]    **Jose Blanchet**[4]    **Vahid Tarokh**[2]

[1]Department of Biomedical Engineering, Duke University, Durham, North Carolina, USA
[2]Department of Electrical and Computer Engineering, Duke University, Durham, North Carolina, USA
[3]Instituto de Matemática Pura e Aplicada, Rio de Janeiro, Brazil
[4]Department of Management Science and Engineering, Stanford University, Stanford, California, USA

## Abstract

We propose a neural network architecture that enables non-parametric calibration and generation of multivariate extreme value distributions (MEVs). MEVs arise from Extreme Value Theory (EVT) as the necessary class of models when extrapolating a distributional fit over large spatial and temporal scales based on data observed in intermediate scales. In turn, EVT dictates that $d$-max-decreasing, a stronger form of convexity, is an essential shape constraint in the characterization of MEVs. As far as we know, our proposed architecture provides the first class of non-parametric estimators for MEVs that preserve these essential shape constraints. We show that the architecture approximates the dependence structure encoded by MEVs at parametric rate. Moreover, we present a new method for sampling high-dimensional MEVs using a generative model. We demonstrate our methodology on a wide range of experimental settings, ranging from environmental sciences to financial mathematics and verify that the structural properties of MEVs are retained compared to existing methods.

## 1 INTRODUCTION

Modeling the occurrence of extreme events is an important task in many disciplines such as medicine, environmental science, engineering, and finance. For example, understanding the probability of a patient having an adverse reaction to medication or the distribution of economic shocks is critical to mitigating the associated effects of these events [Dey and Yan, 2016]. However, these events are rare in occurrence and therefore are often difficult to characterize with traditional statistical tools. This has been the primary focus of extreme value theory (EVT), which describes how to ex-

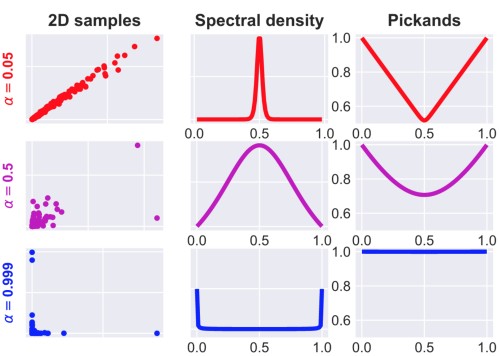

Figure 1: Equivalent representations of MEVs in dimension two, from dependent at the top row to independent at the bottom row. Left column, samples from MEV; Middle column, spectral representation; Right column, Pickands dependence function. We propose methods for estimating the Pickands function (section 2.1), recovering the spectral density (section 2.2) and sampling MEVs (section 4).

trapolate the occurrence of rare events outside the range of available data. In the one-dimensional case, EVT provides remarkably simple models for the asymptotic distribution of the maximum of an infinite number of independent and identically distributed (i.i.d.) random variables, which is due to the celebrated Fisher-Tippet-Gnedenko theorem [Embrechts et al., 1997]. These are known as the generalized extreme value (GEV) distributions [de Haan and Ferreira, 2010].

Perhaps more relevant to practical use-cases is to consider simultaneous extremes in the multi-dimensional scenario. For example, how are extreme weather patterns related in geographical areas or how do extremes of different financial instruments relate? Unlike the one-dimensional case, multivariate extreme value (MEV) distributions generally do not endow simple analytical forms of the underlying density. This leads to difficulties in performing inference tasks using conventional methods. Instead, MEV distributions are characterized by tail dependence functions embedded

*Accepted for the 38th Conference on Uncertainty in Artificial Intelligence* (UAI 2022).

in extreme value copulas [Pickands, 1981, Gudendorf and Segers, 2010].

## BACKGROUND: EXTREME VALUE COPULAS

We start with a brief overview of multivariate EVT and provide additional background material in Appendix A. Let $\Delta_{d-1}$ denote the unit $d-$dimensional simplex. Let $X_i = (X_1^{(i)}, \ldots, X_d^{(i)}) \in \mathbb{R}^d$ for $i \in \{1, \ldots, n\}$ be a sample of i.i.d. random vectors with common continuous probability distribution $F$, marginals $F_1, \ldots, F_d$ and copula $C_F$. The copula $C_F : [0,1]^d \to [0,1]$ is a function that satisfies:

$$C_F(\mathbf{u}) = \mathbb{P}\left[F_1(X_1) \le u_1, \ldots, F_d(X_d) \le u_d\right].$$

Let the vector of *component-wise maxima* be given by: $M^{(n)} = \left(M_1^{(n)}, \ldots, M_d^{(n)}\right)$, where $M_k^{(n)} = \max_{i=1,\ldots,n} X_k^{(i)}$ for $k \in \{1, \ldots, d\}$. Let $C_n$ be the copula of $\bar{M}^{(n)}$ given by: $\bar{M}^{(n)} = \left(\frac{M_1^{(n)} - b_1^{(n)}}{a_1^{(n)}}, \ldots, \frac{M_d^{(n)} - b_d^{(n)}}{a_d^{(n)}}\right)$, where each component-wise maxima $M_k^{(n)}$ is normalized with sequences of real numbers $a_k^{(n)} > 0$ and $b_k^{(n)}$ such that the corresponding limiting marginal is non-degenerate. Then the following property known as *max-stability* holds:

$$C_n(u_1, \ldots, u_d) = C_F(u_1^{1/n}, \ldots, u_d^{1/n})^n, \ \forall \ \mathbf{u} \in [0,1]^d.$$

We are interested in finding the limiting copula $C$ of $C_n$ as $n \to \infty$. The limiting copula is then called an *extreme value copula* and we say that $C_F$ is in the *maximum domain of attraction* of $C$, denoted as $C_F \in \mathrm{MDA}(C)$. The limiting extreme value copula $C$ has the form [Segers, 2012]:

$$C(\mathbf{u}) = \exp\left[\left(\sum_{k=1}^d \log u_k\right) A\left(\frac{\log u_1}{\sum_{k=1}^d \log u_k}, \ldots, \frac{\log u_d}{\sum_{k=1}^d \log u_k}\right)\right], \tag{1}$$

where $A$ is known as a *Pickands dependence function* that defines the joint dependence of a MEV.

**Definition 1** (Pickands dependence function). *A function $A : \Delta_{d-1} \to [1/d, 1]$ is called a Pickands dependence function if it satisfies the following properties:*

1. *$A$ is homogeneous of order 1 and d-max-decreasing where d is the dimension;*

2. *$A$ satisfies $\max_{k=1,\ldots,d} w_k \le A(\mathbf{w}) \le 1$ for all $\mathbf{w} \in \Delta_{d-1}$.*

3. *$A(\mathbf{e}_k) = 1$ where $\mathbf{e}_k$ is the $k^{th}$ canonical basis vector.*

We give the functional definition of *d-max-decreasing* in Appendix B[1] and instead give the spectral correspondence of $A$ here.

**Definition 2** (Spectral form of Pickands dependence function). *For any Pickands dependence function $A$, there exists a Borel measure (spectral measure) $\Lambda$ on $\Delta_{d-1}$ satisfying $\int_{\Delta_{d-1}} s_k \, d\Lambda(\mathbf{s}) = 1$ for $k \in \{1, \ldots, d\}$ such that*

$$A(\mathbf{w}) = \int_{\Delta_{d-1}} \max_{k=1,\ldots,d} w_k s_k \, d\Lambda(\mathbf{s}), \ \mathbf{w} \in \Delta_{d-1}. \tag{2}$$

The equality $\int_{\Delta_{d-1}} s_k \, d\Lambda(\mathbf{s}) = 1$ is only used as a convention to standardize the margins, and is not essential in maintaining the $d$-max decreasing property [Fougères et al., 2013]. To provide some intuition on the aims of this paper, Figure 1 illustrates the relationship between different equivalent representations for a canonical parametric MEV – the symmetric logistic distribution with dependence parameter $\alpha = 0.05$ leaning towards complete dependence and $\alpha = 0.999$ leaning towards complete independence. The proposed methods estimates the Pickands function (right most column) and recovers the spectral measure (middle column) which enables sampling MEVs (left most column).

**Related Work.** A number of techniques have been developed to estimate extreme value copulas from data. The most relevant to the present work is that by Pickands [1981] where a non-parametric estimator of the Pickands function was first proposed. Following works such as Capéraà et al. [1997] and Bücher et al. [2011] describe alternative takes on estimating the dependence function. The above methods, however, do not guarantee that the estimate completely satisfy the conditions of a valid Pickands dependence function. In Marcon et al. [2017], the authors consider a projection of a nonparametric estimator to a convex function represented as a Bernstein polynomial. However, the number of parameters required significantly increases with both the amount of data and the dimensionality, making it difficult for higher dimensional problems or problems with many data points. Finally, a number of Pickands estimators were compared and described in Vettori et al. [2018], and notably none of the estimators reviewed satisfied all requirements of the Pickands function in cases where $d > 2$. For additional details, please refer to the review on extreme value copulas in Gudendorf and Segers [2012]. A theoretical review of $d$-max-decreasing functions and their applications to copulas is given in Ressel [2019].

**Our Contributions.**

1. We present $d$-max-decreasing neural networks, an ar-

---

[1]Intuitively, $d$-max-decreasing describes a stronger form of convexity needed to ensure that subsets of margins remain valid MEVs. See Hofmann [2009, Theorem 5.2.2] and Ressel [2013, Theorem 6] for further details.

chitecture constrained to represent Pickands dependence functions of MEVs.

2. We prove that, in the limit, the proposed architecture can approximate arbitrary Pickands functions.

3. We propose a generative neural network representation of the spectral density of Pickands functions.

4. We propose an extension of the Pickands estimator to train neural networks.

## 2 NEURAL REPRESENTATIONS OF EXTREME VALUE DISTRIBUTIONS

Our main results propose two architectures for representing MEVs: a deterministic method for representing the Pickands dependence function, and a stochastic method for representing the spectral measure. While both represent equivalent quantities, each is more suited for a particular task. The deterministic representation is more suitable for estimating exceedance probabilities whereas the spectral representation is more suitable for sample generation.

### 2.1 D-MAX-DECREASING NEURAL NETWORKS

We are interested in finding a flexible parameterization of $A$ that enforces all the properties given in Definition 1. The most difficult property to enforce is being $d$-max-decreasing. To that end, we propose a new architecture inspired by Maxout Networks [Goodfellow et al., 2013] and Input Convex Neural Networks (ICNNs) [Amos et al., 2017]. The proposed architecture, dubbed $d$-max Neural Networks (dMNNs), has additional restrictions to fulfill the conditions of the Pickands dependence function.

**Theorem 1** ($d$-max-decreasing Neural Architecture). *Let $A_{\boldsymbol{\theta}}^{(m)}(\mathbf{w})$ be a function defined as:*

$$A_{\boldsymbol{\theta}}^{(m)}(\mathbf{w})$$
$$:= \max\left(\max_{k=1,\dots,d} w_k, \; L^{(m)}(\mathbf{w}) + (1 - L^{(m)}(\mathbf{e})^T \mathbf{w})\right), \tag{3}$$

*where*

$$L^{(m)}(\mathbf{w}) = \frac{1}{n_m}\sum_{j=1}^{n_m}\left(\ell^{(m)} \circ \ell^{(m-1)} \circ \cdots \circ \ell^{(1)}(\mathbf{w})\right)_j,$$

$$\ell^{(i)}(\mathbf{h}^{(i-1)})_j = \max_{k=1,\dots,n_{i-1}}\left(\Theta_{j,\cdot}^{(i)} \odot h^{(i-1)}\right)_k,$$

$$\mathbf{h}^{(i-1)} = \ell^{(i-1)} \circ \cdots \circ \ell^{(1)}(\mathbf{w}),$$

$$L(\mathbf{e}) = (L(\mathbf{e}_1),\dots,L(\mathbf{e}_d))^T,$$

*$m$ is the number of layers, $n_i$ is the width of the $i^{th}$ layer, $\Theta^{(i)} \in \mathbb{R}_+^{n_i \times n_{i-1}}$ are the weights of the $i^{th}$ layer, constrained*

*to be all positive, and $\mathbf{e}_i$ is the $i^{th}$ canonical basis vector. $\odot$ denotes component-wise multiplication.*

*Then, $A_{\boldsymbol{\theta}}^{(m)}(\mathbf{w})$ is a $d$-max-decreasing function. Moreover, $A_{\boldsymbol{\theta}}^{(m)}(\mathbf{w})$ represents a valid Pickands dependence function.*

*Intuition of proof.* The proof uses the idea that $\mathbb{E}_{\mathbf{s}}[\max_{k=1,\dots,d}(w_k s_k)]$, $\mathbf{s} \in \Delta_{d-1}$ is $d$-max-decreasing and certain compositions of this function retain this property. The full proof is given in Appendix C.1. □

For notational convenience, we drop the $(m)$ unless needed. To get an intuition behind the structure of the architecture, note that in the single layer case in the limit as $n_1 \to \infty$, the weights $\boldsymbol{\theta}$ correspond to samples of the spectral measure in Definition 2 and the expectation is computed empirically. While the proposed architecture is guaranteed to enforce the properties of the Pickands function, and is thus $d$-max-decreasing, we are also interested in seeing how well it can approximate an arbitrary Pickands dependence function. We present results in the following theorem:

**Theorem 2** (Uniform Convergence). *Suppose that $\boldsymbol{\theta}$ are samples from the true spectral measure and $A$ is the true Pickands function. The empirical process*

$$\mathbb{G}_n = \sqrt{n}\left(A_{\boldsymbol{\theta}}^{(1)}(\mathbf{w}) - A(\mathbf{w})\right)$$

*converges to a zero mean Gaussian process as $n \to \infty$ where $A_{\boldsymbol{\theta}}^{(1)}$ is a single layer dMNN of width $n$.*

*Intuition of proof.* We first establish pointwise convergence. Then we show $A$ is Lipschitz over a bounded set whose covering number grows in accordance with functions that are $P-$Donsker. The full proof is given in Appendix C.2. □

The result in Theorem 2 has many implications on the properties of the proposed network since it, for example, allows us to quantify the uncertainty associated with our function estimates. Using the proposed architecture, we mitigate issues faced by previous estimators, such as [Bücher et al., 2011, Capéraà et al., 1997, Marcon et al., 2017], in enforcing the $d$-max-decreasing property, inequalities, and endpoints of the function.

### 2.2 A GENERATIVE MODEL FOR THE SPECTRAL MEASURE

While the spectral measure can be computed from the weights of the proposed dMNN, we propose an alternative representation of the spectral measure using a generative neural network. We model $\mathbf{y} \sim \Lambda$ in (2) as the output of a generative neural network $G(\,\cdot\,;\boldsymbol{\phi}) \in \mathbb{R}_+^d$ with parameters $\boldsymbol{\phi}$, i.e. $\mathbf{y} = G(\mathbf{z};\boldsymbol{\phi})$ which maps input samples $\mathbf{z} \sim p_z$ to $\mathbf{y}$,

where $p_z$ is a distribution that is easy to sample from (such as a multivariate Gaussian distribution). This leads us to a representation of $A$ in terms of the generator:

$$A_G(\mathbf{w}) := \mathbb{E}_{\mathbf{y} \sim G} \left[ \max_{k=1,\ldots,d} w_k y_k \right], \qquad (4)$$

where $\mathbb{E}[y_k] = 1$. The expectation is taken empirically with a large number of samples from $G$.

**Remark 1.** *The function given by* (4) *satisfies all the necessary conditions for a valid Pickands function.*

Following Remark 1, we informally note that it follows from the universal approximation theorem of neural networks that if $G$ is sufficiently expressive then (4) can represent an arbitrary Pickands dependence function.

**Use Cases of Each Representation.** The difference between the representation given by the $d$MNN (3) and the generative neural network (4) is: in the $d$MNN case the spectral measure is modeled by a discrete number of elements as dictated by the $d$MNN architecture, while in the generator case the implicit distribution of the spectral measure is modeled. The $d$MNN is useful in representing probabilistic quantities since it provides a deterministic representation of the CDF and therefore it does not exhibit the variance of the generative representation. On the other hand, the generative model is capable of simulating many realizations of the MEV, particularly useful for sampling applications.

# 3 PARAMETER ESTIMATION

Fitting data to high dimensional copulas is often a difficult task since the probability density function (PDF) is not directly modeled. In general, specific parametric families are used to make the process easier, such as in Archimedean copulas. While it is theoretically possible to first obtain the underlying PDF via differentiating the CDF and then fit the $d$MNN with Maximum Likelihood Estimation (MLE), the procedure is computationally complex, especially in high dimensions. The main drawback of such a method lies in the need to differentiate the $d-$variate CDF, since nested differentiation with existing automatic differentiation methods may result in numerical errors [Margossian, 2019]. Instead, we use specific properties of MEVs to transform the parameter fitting procedure into MLE over univariate random variables. We additionally present the analogs for survival distributions in Appendix D.

## 3.1 FITTING THE DEPENDENCE FUNCTION

Let $F_k$ denote the univariate marginal CDF (which can be fitted using MLE as in Embrechts et al. [1997] or the $L$-moments method of Hosking [1990]) of the $k^{\text{th}}$ normalized component wise maxima $\bar{M}_k^{(n)} = \frac{M_k^{(n)} - b_k^{(n)}}{a_k^{(n)}}$,

$k \in \{1,\ldots,d\}$. In addition, let $\mathbf{w} = (w_1,\ldots,w_d) \in \Delta_{d-1}$. We introduce the transformation on $\bar{M}_k^{(n)}$:

$$\widetilde{M}_k^{(n)} = -\log(F_k(\bar{M}_k^{(n)})), \ \forall k \in \{1,\ldots,d\}, \quad (5)$$

$$Z_w = \min_{k=1,\ldots,d} \widetilde{M}_k^{(n)} / w_k. \qquad (6)$$

Then, we have: $\mathbb{P}\left[ Z_w > z \right] = e^{-zA(\mathbf{w})}$ (for the full derivation, see Section 3 of Gudendorf and Segers [2012]). This transformation casts the original multi-dimensional distribution into the new variables $Z_w$ that are exponentially distributed with rate parameter given by the Pickands dependence function $A(\mathbf{w})$. From this transformation, we can fit the model $A_{\boldsymbol{\theta}}(\mathbf{w})$ to samples $Z_w$ using MLE. This can be done by training the model $A_{\boldsymbol{\theta}}(\mathbf{w})$ with stochastic gradient descent (SGD) to match the data points $Z_w$ as follows:

$$A_{\boldsymbol{\theta}}^\star(\mathbf{w}) = \arg\min_{\boldsymbol{\theta}} \mathbb{E}_{Z_w} \mathcal{L}(Z_w; \boldsymbol{\theta}), \qquad (7)$$

where

$$\mathcal{L}(Z_w; \boldsymbol{\theta}) = A_{\boldsymbol{\theta}}(\mathbf{w}) Z_w - \log A_{\boldsymbol{\theta}}(\mathbf{w}). \qquad (8)$$

Alternative losses could be considered by reformulating the loss with respect to the estimators defined in Bücher et al. [2011] and Capéraà et al. [1997]. We empirically found that the MLE approach described in (7) provides the best performance, and it follows naturally from the original formulation of Pickands [1981]. The training procedure is summarized in Algorithm 1.

---

**Algorithm 1** Fitting the Pickands-$d$MNN to Data

---

1: **Input:** $\left\{ \left( X_1^{(i)}, \ldots, X_d^{(i)} \right) \right\}_{i=1}^N$, $N = B \times n$ samples of i.i.d. random vectors where $B$ is the number of blocks of data and $n$ is the size of each block.
2: Take component-wise maxima over each block: $\left\{ \left( M_1^{(n,b)}, \ldots, M_d^{(n,b)} \right) \right\}_{b=1}^B$ where
$$M_k^{(n,b)} = \max_{i=(b-1)n+1,\ldots,bn} X_k^{(i)},$$
for $k \in \{1,\ldots,d\}$ and $b \in \{1,\ldots,B\}$.
3: Fit a GEV to each component-wise maxima $\{M_k^{(n,b)}\}_{b=1}^B$, obtain $\{\bar{M}_k^{(n,b)}\}_{b=1}^B$, then estimate marginals $F_k$ for each $k \in \{1,\ldots,d\}$.
4: **Initialize** the parameters $\boldsymbol{\theta} \geq 0$ of the $d$MNN
   **Repeat**:
5: Randomly sample a minibatch of training data $\{\bar{M}_k^{(n,b)}\}_{b\in\text{batch}}$ and uniformly sample $\mathbf{w} \in \Delta_{d-1}$.
6: Transform samples according to Equations (5) and (6) to obtain transformed samples $\{Z_{w,b}\}_{b\in\text{batch}}$.
7: Compute gradient $\nabla_{\boldsymbol{\theta}} \sum_{b\in\text{batch}} \mathcal{L}(Z_{w,b}; \boldsymbol{\theta})$.
8: Update $\boldsymbol{\theta}$ with Adam [Kingma and Ba, 2014].
   **Until** convergence
   **Output:** $A_{\boldsymbol{\theta}}^\star(\mathbf{w})$.

---

## 3.2 FITTING THE GENERATOR

Recall that we have an equivalent representation of $A$ given by $A_G$ in (4) where $G(\cdot; \phi)$ is a function, with parameters $\phi$, of random variables. We fit the parameters $\phi$ of the generator by solving the following optimization problem:

$$\min_{\phi} \mathbb{E}_{Z_w} \mathcal{L}(Z_w; \phi) + \eta \left\| \mathbb{E}_{\mathbf{y}}[\mathbf{y}] - \mathbf{1}_d \right\|_2^2, \qquad (9)$$

with $\mathcal{L}$ now defined using the representation of $A_G$ in (4):

$$\mathcal{L}(Z_w; \phi) = \mathbb{E}_{\mathbf{y}} \big[ \max_{k=1\ldots d} y_k w_k \big] Z_w - \log \mathbb{E}_{\mathbf{y}} \big[ \max_{k=1\ldots d} y_k w_k \big],$$

where $\mathbf{y} = (y_1, \ldots, y_d) = G(\mathbf{z}; \phi)$, $\mathbf{y} \in \mathbb{R}_+^d$ and $\mathbf{z} \in \mathbb{R}^k \sim p_z$ with $\eta > 0$ as a regularization factor. Note that the second expectation in (9) is only needed to enforce the margins. It need not be strictly enforced, enforcing approximately only results in minor changes in the tail index. The expectations with respect to $\mathbf{y}$ in (9) are approximated using the sample mean with samples from the generator.

To summarize the parameter estimation section, we bypass the need to differentiate the CDF and use properties of MEVs to estimate the parameters of the distribution from data. Both representations of the Pickands function presented can be used with this technique.

## 4 SAMPLING

While learning MEV distributions from data is important for computing probabilities, it is also useful to simulate possible scenarios by sampling from an estimated MEV distribution. We introduce a sampling technique using the proposed architectures to efficiently sample from arbitrary MEVs. To the best of our knowledge, there are no general sampling methods for arbitrary extreme value copula that scale to high dimensions. This is because MEV sampling algorithms assume knowledge of the spectral measure, and do not consider sampling when given only the Pickands function. It then becomes necessary to recover the spectral measure from a given Pickands function or from data, which we previously described two methods for doing so. We additionally note that the traditional method of conditional sampling for copulas is ineffective since it requires both computing high order derivatives and using numerical root-finding techniques. We base our sampling procedure on algorithms for the infinite dimensional analogue of MEV distributions known as *max-stable processes* [Dombry et al., 2016]. Max-stable processes have the property that finite dimensional marginals are MEVs and have a spectral representation in terms of the spectral measure $\Lambda$ for stationary processes. This ultimately allows us to recast MEV sampling in terms of prior work on sampling from max-stable processes, where established methods exist.

## 4.1 MARGINS OF MAX-STABLE PROCESSES AS MEV DISTRIBUTIONS

A stationary max-stable process has the form:

$$\max_{i \geq 1} \xi_i y_i(x), \ \ x \in \mathbb{X} \subset \mathbb{R}^k \qquad (10)$$

where $\xi_i$ is the $i^{\text{th}}$ realization of a Poisson point process with intensity $\xi^{-2} \mathrm{d}\xi$. $y_i$ is the $i^{\text{th}}$ sample from the spectral measure. Additionally, $\mathbb{E}[y(x)] = 1$, $x \in \mathbb{X}$ is generally assumed to enforce unit Frechet margins. For a finite number $d$ of $\{x_j\}_{j=1}^d$, this corresponds to a $d$-dimensional spectral measure with the same properties as in Definition 2. The key idea is to use the representation in (10) to sample from the full MEV distribution with only knowledge of the spectral measure. We use the algorithm mentioned in Hofert et al. [2018, Algorithm 1] for sampling from the full distribution given samples of the spectral measure. We give the details of the algorithm in Appendix J Algorithm 4.

## 4.2 SAMPLING FROM THE DMNN

Suppose we fit a single layer $d$MNN using Algorithm 1 with weights given by $\theta \in \mathbb{R}_+^{w \times d}$ where $w$ is the width of the network and $d$ is the data dimension. Consider the transformation $\hat{\theta}_{i,j} = \theta_{i,j} / \sum_{j=1}^d \theta_{i,j}$ where we transform the weights of the network to the unit simplex $\Delta_{d-1}$, and $i, j$ refer to the row and column indices.

We then choose a number $N$ and compute

$$\max_{i=1,\ldots,N} \xi_i \hat{\theta}_{i+j}, \quad j \sim \mathrm{rand}(\{1, \ldots, w - N\})$$

where $\xi_i$ is defined as per (10). While this method is effective in sampling, a possible issue is the finite number of $\hat{\theta}$ dictated by the width $w$ of the network. The generative model on the other hand allows for unlimited generation of samples of the spectral measure.

## 4.3 SAMPLING FROM THE GENERATIVE MODEL

Suppose we fit a generative model $G(z; \phi)$ to data following the optimization procedure in (9). Then sampling proceeds similarly to the case with the $d$MNN except in this case we do not use the weights of the network explicitly, but sample from the model:

$$\max_{i=1,\ldots,N} \xi_i y_i \text{ where } y_i = G(z_i; \phi), z_i \sim p(z)$$

where the notation is maintained as above with $p(z)$ defining an easy to sample prior distribution.

As a final note regarding the sampling methods, one particularly useful way of combining the methods is to first estimate $A_\theta$ from data using an estimator such as the $d$MNN. Then,

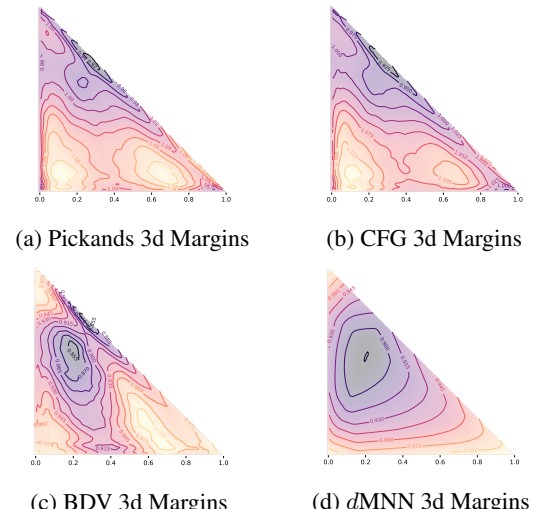

(a) Pickands 3d Margins     (b) CFG 3d Margins

(c) BDV 3d Margins     (d) $d$MNN 3d Margins

Figure 2: Qualitative comparison of 3d margins from learned 10d MEV for the commodities dataset. The $d$MNN retains margins that are valid Pickands dependence function. The other estimators are non-convex and outside the required bounds. Contours plotted with solid lines. See additional figures in Appendix I and E, Figures 14d to 17d.

fit the generator to $A_{\boldsymbol{\theta}}$ by taking the mean squared error (MSE) between the two representations, i.e.

$$\min_{\boldsymbol{\phi}} \mathbb{E}_{\boldsymbol{w} \sim \mathrm{Unif}(\Delta_{d-1})}(A_{\boldsymbol{\theta}}(\boldsymbol{w}) - A_G(\boldsymbol{w}))^2$$
$$+ \eta \|\mathbb{E}[\mathbf{y}] - \mathbf{1}_d\|_2^2.$$

This provides a simple way to recover the spectral density of any given EVC and thus an effective way to sample from arbitrary MEVs. We detail this algorithm in Appendix J Algorithm 3.

## 5 RESULTS

In this section, we provide numerical results that compare the estimation capabilities of the proposed $d$MNN-based model with well-known estimators from the literature: Pickands [Pickands, 1981], CFG [Capéraà et al., 1997], and the estimator described in [Bücher et al., 2011] which we refer to as BDV. These estimators are described in greater detail in Appendix G. We start by evaluating the performance for estimating survival probabilities on known parametric models, followed by real data. We conclude with experiments on sampling from a MEV, where we use the proposed generative model for high dimensional data with different dependence structures. To align with the results in Theorem 2, for the experiments presented in this section, we use a single layer $d$MNN with a width of 512. Additional experiments with two different architectures are presented in Appendix E. Code for experiments is available at[2].

[2] https://github.com/alluly/dMNN

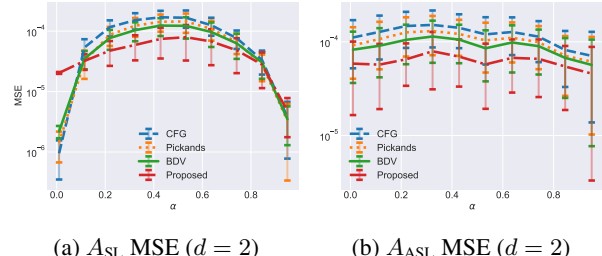

(a) $A_{\mathrm{SL}}$ MSE ($d = 2$)     (b) $A_{\mathrm{ASL}}$ MSE ($d = 2$)

Figure 3: MSE of survival probabilities for $d = 2$ with 100 samples for $A_{\mathrm{SL}}$ (3a) and $A_{\mathrm{ASL}}$ (3b). Thresholds are above the 75th percentile.

| Pickands function | Parameters |
|---|---|
| $A_{\mathrm{SL}}(\mathbf{w}) = \left(\sum_{k=1}^{d} w_k^{1/\alpha}\right)^{\alpha}$ | $\alpha \in (0, 1]$ |
| $A_{\mathrm{ASL}}(\mathbf{w}) = \sum_{b \in \mathcal{P}_d} \left(\sum_{i \in b}(\lambda_{i,b}w_i)^{1/\alpha_b}\right)^{\alpha_b}$ | $\alpha_b \in (0, 1]$ $\lambda_{i,b} \in [0, 1]$ $\sum_{i \in b} \lambda_{i,b} = 1$ |

Table 1: Parametric Pickands functions for the symmetric $A_{\mathrm{SL}}$ and asymmetric $A_{\mathrm{ASL}}$ logistic copulas and their valid parameter ranges. $\mathcal{P}_d$ refers to the power set of $\{1, \ldots, d\}$. All functions are defined for domain $\mathbf{w} \in \Delta_{d-1}$.

**Synthetic data.** We consider two canonical families of extreme value distributions known as the symmetric logistic ($A_{\mathrm{SL}}$) and the asymmetric logistic ($A_{\mathrm{ASL}}$) families where the underlying Pickands function is given by Gudendorf and Segers [2010] listed in Table 1. $\alpha \in (0, 1]$ is the parameter modeling the degree of dependence between variables ranging from complete dependence ($\alpha = 0$) to complete independence ($\alpha = 1$). Exact sampling from distributions of this type are described in Stephenson [2003]. Note that for both the symmetric and asymmetric copulas, the marginals are distributed according to the standard Fréchet distribution. We start by comparing the MSE of survival probabilities for $d = 2$ where the true Pickands dependence function is given by the symmetric or asymmetric model described above for different degrees of dependence $\alpha$. We compute the exact values of the survival probability and consider survival probabilities associated with margins above the 75th percentile. As shown in Figures 3a and 3b, the proposed Pickands-$d$MNN estimator achieves the lowest MSE performance for most degrees of dependence $\alpha$ for the symmetric logistic model and all the degrees of the asymmetric logistic model. The proposed method performs worse comparatively in the full dependence case of the symmetric logistic (when all components of the vector are the same) which we suspect is due to difficulties in the optimization procedure of the $d$MNN. We additionally showcase the ability of the proposed method to model high dimensional extreme value distributions. To do this, we train the Pickands-$d$MNN with data for $d = 256$ with $\alpha \in \{0.25, 0.50, 0.75, 1.0\}$ and for $d = \{256, 512, 728, 1024\}$ with $\alpha = 0.5$. Then, we com-

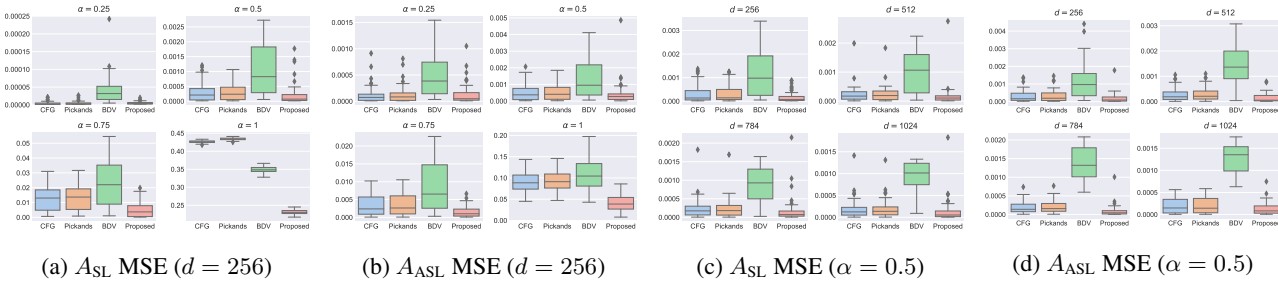

(a) $A_{\text{SL}}$ MSE ($d = 256$)  (b) $A_{\text{ASL}}$ MSE ($d = 256$)  (c) $A_{\text{SL}}$ MSE ($\alpha = 0.5$)  (d) $A_{\text{ASL}}$ MSE ($\alpha = 0.5$)

Figure 4: Comparison of $||\hat{A}(\mathbf{w}) - A(\mathbf{w})||_2^2$ for different estimators $\hat{A}$ for different dependence $\alpha = \{0.25, 0.50, 0.75, 1.0\}$ with fixed $d = 256$ (4a, 4b) and for fixed $\alpha = 0.5$ with different $d = \{256, 512, 728, 1024\}$ (4c, 4d). The reference $A(\mathbf{w})$ are $A_{\text{SL}}$ (4a, 4c) and $A_{\text{ASL}}$ (4b, 4d). Results are over 50 runs with 100 training samples for each run.

pute the MSE between the Pickands-$d$MNN and the true Pickands function via Monte Carlo with 10,000 uniformly sampled points in $\Delta_{d-1}$. The results are illustrated for varying $\alpha$ in Figures 4a and 4b and for $\alpha = 0.5$ in Figures 4c and 4d. While all hyperparameters were fixed at the beginning and not fine-tuned, we note that performance may improve if additional fine-tuning is performed using a validation set.

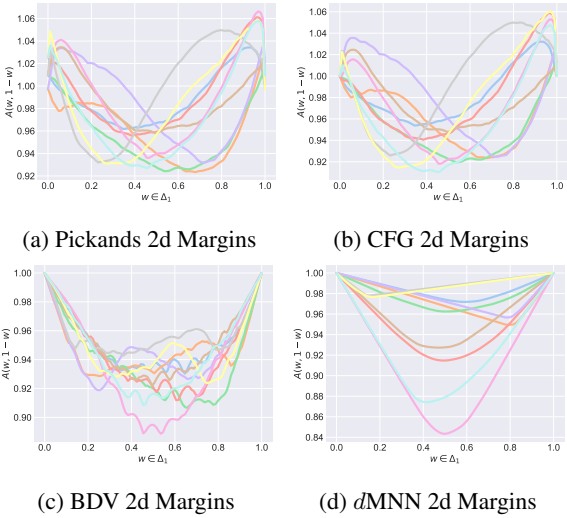

(a) Pickands 2d Margins  (b) CFG 2d Margins

(c) BDV 2d Margins  (d) $d$MNN 2d Margins

Figure 5: Qualitative comparison of 10 out of 45 total 2d margins from learned 10d MEV for the California Winds dataset. The $d$MNN is the only method that retains margins that are valid Pickands dependence functions. See additional figures in Appendix I and E. Figures 14d - 17d.

**Real data.** We test the proposed estimator with real data on extreme ozone levels ($d = 4$), wind gusts ($d = 10$), commodity prices ($d = 10$), cryptocurrencies to USD conversion rates ($d = 100$), S&P 500 components with sufficient history ($d = 418$), and county-level COVID-19 case counts for California ($d = 58$), New York ($d = 58$) and North Carolina ($d = 100$). We provide details for each dataset in Appendix H. For environmental datasets, we compute the maximum over the different sampling periods, while for the

financial data we compute the maximum drawdown. The maximum drawdown is defined as the difference between the minimum and maximum values over a time period normalized by the maximum value. For the COVID-19 data, we compute the change in case counts over different time scales. All margins were fit with GEVs using the `scipy` implementation, which computes the $a_n, b_n$ normalizing constants.

The main challenge associated with real data is the lack of a ground truth for comparison purposes. It is extremely difficult to accurately compare different estimators on real data because we can never observe the true distribution of extremes. Since the purpose of EVT is to extrapolate to the tails from observations not necessarily in the tails, we consider extreme events on different time scales. If we fit based on extreme observations on shorter time scales and test on extreme observations on longer time scales, we will obtain an estimate of how well the different methods extrapolate to tail probabilities, since longer time scales will have more extreme events.

We compute the accuracy of the different estimators with respect to the empirical estimate on held out data over longer time scales. Specifically, we choose a series of quantiles where we observe data and compute the difference between the estimated survival probabilities and the empirical estimate calculated from observed data. This is quantified as: $\frac{1}{|Q|} \sum_{\gamma \in Q} \left[ \frac{1}{B} \sum_{b=1}^{B} \mathbb{1}_{\{M_{n,b} \geq \gamma\}} - P_\theta(M_n \geq \gamma) \right]^2$, where $M_{n,b} = \left( M_{n,b}^{(1)}, \ldots, M_{n,b}^{(d)} \right)$ is the $d$-dimensional vector of point-wise maxima (or point-wise maximum drawdown over a period of interest), $P_\theta$ is the estimated survival probability, and $Q$ is a set of thresholds to consider. We choose $Q$ to be all quantiles such that the empirical probability is greater than 0. This measures how well the proposed method can extrapolate to greater extremes over longer time scales. The results are presented in Table 2 and suggest that while most estimators perform similarly, the proposed method most consistently performs the best in terms of the evaluation metric. We would like to emphasize that empirical evaluation on real data is very challenging, and the high variances prevent

| | $d$ | Train/Test | PICKANDS | CFG | BDV | PROPOSED |
|---|---|---|---|---|---|---|
| Wind | 10 | day/week | $4.48(18.6)_{\times 10^{-4}}$ | $4.15(15.1)_{\times 10^{-4}}$ | $\mathbf{4.10(16.3)}_{\times 10^{-4}}$ | $4.37(17.5)_{\times 10^{-4}}$ |
| Ozone | 4 | day/week | $3.06(4.66)_{\times 10^{-2}}$ | $2.99(4.56)_{\times 10^{-2}}$ | $2.86(4.46)_{\times 10^{-2}}$ | $\mathbf{2.73(4.25)}_{\times 10^{-2}}$ |
| Commodities | 10 | week/month | $4.34(5.82)_{\times 10^{-3}}$ | $4.33(5.71)_{\times 10^{-3}}$ | $1.60(1.96)_{\times 10^{-3}}$ | $\mathbf{1.56(2.21)}_{\times 10^{-3}}$ |
| S&P 500 | 418 | week/month | $3.02(21.2)_{\times 10^{-3}}$ | $3.02(21.1)_{\times 10^{-3}}$ | $6.28(35.2)_{\times 10^{-3}}$ | $\mathbf{2.41(22.2)}_{\times 10^{-3}}$ |
| Crypto | 100 | week/month | $1.06(2.85)_{\times 10^{-2}}$ | $1.05(4.86)_{\times 10^{-2}}$ | $1.34(3.44)_{\times 10^{-2}}$ | $\mathbf{8.57(26.4)}_{\times 10^{-3}}$ |
| COVID (NC) | 100 | week/week | $4.04(7.21)_{\times 10^{-2}}$ | $4.04(7.19)_{\times 10^{-2}}$ | $3.83(6.51)_{\times 10^{-2}}$ | $\mathbf{4.37(10.7)}_{\times 10^{-3}}$ |
| COVID (NY) | 58 | week/week | $2.74(10.4)_{\times 10^{-2}}$ | $2.74(10.4)_{\times 10^{-2}}$ | $2.25(7.75)_{\times 10^{-2}}$ | $\mathbf{4.06(9.50)}_{\times 10^{-3}}$ |
| COVID (CA) | 58 | week/week | $1.17(3.98)_{\times 10^{-2}}$ | $1.19(3.87)_{\times 10^{-2}}$ | $1.17(3.85)_{\times 10^{-2}}$ | $\mathbf{1.18(4.83)}_{\times 10^{-3}}$ |

Table 2: MSE of different estimators in estimating maxima over longer time scales. Best and second best performances are marked in **bold** and *italic* respectively.

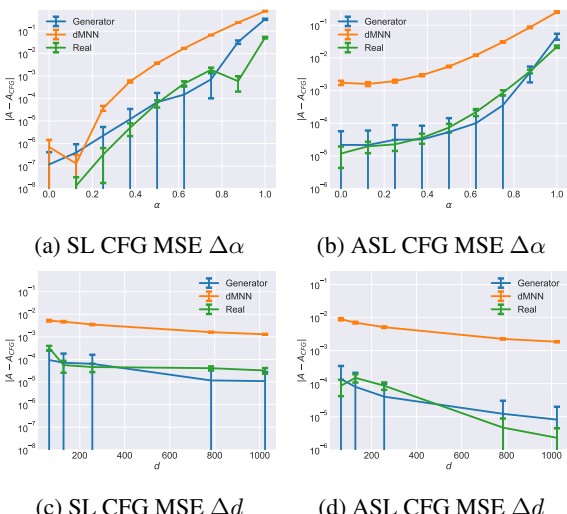

(a) SL CFG MSE $\Delta\alpha$     (b) ASL CFG MSE $\Delta\alpha$

(c) SL CFG MSE $\Delta d$     (d) ASL CFG MSE $\Delta d$

Figure 6: MSE of CFG estimate for 1000 samples and 1000 simplex points for $d = 225$ at various $\alpha \in (0,1)$ (6a, 6b) and for $\alpha = 0.5$ at various $d = \{64, 128, 256, 784, 1024\}$ (6c, 6d). Data sampled from generative model (blue), $d$MNN (orange), and ground truth (green), where the distributions considered were $A_{\text{SL}}$ (6a) and $A_{\text{ASL}}$ (6b). Both models were trained with 1000 data points.

us from making meaningful statements on the efficacy of any of the methods. However, from Figures 2d and 5d, we see that our proposed estimator is the only one that satisfies the necessary properties of the Pickands function, which is the main purpose of the proposed method. Specifically, if we consider the properties of convexity and bounds, the proposed estimator is the only one that retains these. The other estimators are not convex and achieve values greater than 1, which leads to incorrect probabilities when considering conditional probabilities. Additional figures in Appendix I showcase this property on additional datasets and Appendix E Figures 5 to 10 compares these for different architectures. It is critical that these properties are satisfied so that downstream tasks such as conditional probabilities can be computed. From the state-of-the-art estimators, the

properties are not satisfied and thus the applicability of the estimators is severely limited.

**Conditional Prediction.** One important task is computing the conditional survival probability of a random variable. Suppose we have a $d$-dimensional EVC and we are interested in computing the probability that the $i^{\text{th}}$ component exceeds a threshold conditioned on some subset of the other components, $\mathcal{C} \subseteq \{1, \dots, d\}/i$. We can compute this through the relation:

$$\mathbb{P}(x_i > X_i | \cap_{j \in \mathcal{C}} x_j > X_j) = \frac{\mathbb{P}(x_i > X_i, \cap_{j \in \mathcal{C}} x_j > X_j)}{\mathbb{P}(\cap_{j \in \mathcal{C}} x_j > X_j)}.$$

For performance evaluation, we can cast this as a classification problem where we consider features $X_j$, $j \in \mathcal{C}$ with a positive class associated if the combination of $\{X_i, X_j\}, j \in \mathcal{C}$ appears in the held out data. Therefore we only have examples of positive classes since all the examples in the held out data are realizations that did occur. Since we cannot observe the examples that do not occur, we must evaluate how well the method is performing based on the examples that do. This classification problem with a single class was studied in Lee and Liu [2003], where they propose a metric that behaves similarly to the F1 score in binary classification. This metric is defined as

$$\frac{r^2}{\mathbb{E}[\mathbb{1}\{\mathbb{P}(x_i > X_i | \cap_{j \in \mathcal{C}} x_j > X_j) \geq 0.5\}]}, \quad (11)$$

where $r = \frac{1}{N} \sum_{k=1}^{N} \mathbb{1}\{\mathbb{P}(x_i^{(k)} > X_i^{(k)} | \cap_{j \in \mathcal{C}} x_j^{(k)} > X_j^{(k)}) \geq 0.5\}$ is the proportion of correctly classified examples on the held out data. The denominator is approximated by taking an empirical average over the space $[0,1]^{|\mathcal{C}|+1}$. The score has a range of $[0, \infty)$ where larger values indicate better performance. Table 3 presents the results of classification on held out data for the COVID datasets where the top 5 most populous counties are used to predict the probability of the $6^{\text{th}}$ county having the change in case counts at or greater than the observed value. We consider greater than or equal to due to case counts often being underestimated due to lack of testing. The results in Table 3 suggest that

the method is an effective tool for computing conditional probabilities necessary for classification tasks.

|           | NC                    | NY                    | CA                    |
|-----------|-----------------------|-----------------------|-----------------------|
| PICKANDS  | $8.31 \times 10^{-1}$ | $9.68 \times 10^{-1}$ | $8.46 \times 10^{-1}$ |
| CFG       | $8.32 \times 10^{-1}$ | $9.69 \times 10^{-1}$ | $8.46 \times 10^{-1}$ |
| BDV       | $8.10 \times 10^{-1}$ | $8.04 \times 10^{-1}$ | $7.50 \times 10^{-1}$ |
| PROPOSED  | $\mathbf{9.79 \times 10^{-1}}$ | $\mathbf{1.08 \times 10^{0}}$ | $\mathbf{1.10 \times 10^{0}}$ |

Table 3: Classification score (11) on held out COVID-19 data for different states conditioned on 5 counties. Higher is better.

**Sampling from the copula.** Finally, to determine the efficacy of sampling from an arbitrary Pickands copula, we consider two synthetic examples using the previously described MEV distributions in Table 1. In this experiment, we train the generator $G(\,\cdot\,; \phi)$ in (9) based on 1000 samples from the target distribution. We represent $G(\,\cdot\,; \phi)$ as a 2 layer 256 width multi-layer perceptron with ReLU activation functions and set $\eta = 1$. Since the Pickands function completely determines the dependency of the random variables, we compare the CFG estimate of the Pickands function from generated samples to the true Pickands function as a measure of sampling quality. We use the CFG estimator due to its ubiquity in the literature and its highly regarded status as a standard estimator for the Pickands dependence function. The results for generating 225 dimensional samples with varying dependence $\alpha \in [0, 1]$ are shown in Figures 6a and 6b. The figures suggest that the generative model performs comparatively well for both distributions considered, with the worst performance occurring in the nearly independent cases ($\alpha = 1$). This is expected, since independence implies a spectral measure with delta functions on the corners of the simplex, which is difficult to learn (see the bottom row of Figure 1 as an example). The figures additionally suggest that sampling using the learned weights of the $d$MNN has lower variance (since the spectral measure in this is a finite discrete approximation) but does not perform as well in sampling as the generative model. The error of the CFG estimate for the proposed sampling methods (blue and orange) and the exact sampling (green) follow very similar trends in errors, suggesting that both sampling methods are recovering the true spectral measure.

## 6   CONCLUDING REMARKS

We introduced a new neural network architecture for modeling MEV distributions while enforcing all the properties of the distribution. We additionally show that the architecture can approximate any Pickands function, which allows for precise representations of MEV distributions. Finally, we present a generative model for recovering the spectral representation. Numerical results are provided to empirically demonstrate the effectiveness of the methods in their respective tasks. However, there are some limitations of the proposed methods.

**Limitations of Pickands-$d$MNNs and Generative Model.** The main challenge associated with modeling using $d$MNNs are optimization and architectural choices. Choosing appropriate hyperparameters is a difficult and opaque task that requires additional care. This is a case where non-parametric methods are advantageous, at the cost of being unable to guarantee the necessary properties of the function. In general, we suggest using a wide architecture with single depth, since this is the architecture that most of the theory builds upon. Additional progress on understanding the training of deep neural networks should improve the representational capabilities of the $d$MNNs, given its theoretical potential to approximate any Pickands functions to arbitrary precision. Optimization of the generative model suffers from the same issues. Furthermore, since the proposed method requires training a neural network for estimation, the non-parametric methods have a significant computational advantage. In practice, this is not a major issue since these estimators are generally fit once and the proposed method takes only a few seconds on a GPU to fit.

**Future Work.** The proposed methods have possible applications in a variety of modeling situations. One possibility is to extend the application on estimating conditional probabilities of $d$MNN for other classification tasks, such as out of distribution detection. Another is in using the spectral measure for finding groups of variables that are extreme simultaneously, such as in [Engelke and Ivanovs, 2021]. Finally, applications of extremes are important in understanding robustness properties of neural networks [Weng et al., 2018], and the proposed work provides foundation for high dimensional extensions.

## Acknowledgements

Material in this paper is based upon work supported by the Air Force Office of Scientific Research under award number FA9550-20-1-0397. AH was supported by NSF Graduate Research Fellowship.

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
