# OpenReview forum: "Modeling Extremes with $d$-max-decreasing Neural Networks"
_auai.org/UAI/2022/Conference — UAI 2022 Oral_

### Official Review · Reviewer_Z1k7 · 2022-04-11

**Q2(1) Originality/Novelty:** 3
**Q2(2) Significance/Impact:** 3
**Q2(3) Correctness/Technical Quality:** 3
**Q2(6) Clarity Of Writing:** 3
**Q6 Overall Score:** 7
**Q8 Confidence In Your Score:** 2

**Q1 Summary And Contributions:**

The paper proposes a new method for estimating the Pickands function and recovering the spectral density using neural networks. The method for representing the Pickands dependence function is introduced as dMNNs and the method for generating the spectral density uses a generative model formulation.

**Q2 Assessment Of The Paper:**

More detailed information regarding each of these aspects is given below:

**Q2(4) Quality Of Experiments (Optional):**

3: Good: The experimental evaluation is adequate, and the results convincingly support the main claims.

**Q2(5) Reproducibility:**

3: Good: Key resources (e.g., proofs, code, data) are available and key details (e.g., proofs, experimental setup) are sufficiently well-described for competent researchers to confidently reproduce the main results.

**Q3 Main Strengths:**

* The motivation for this work is important. Combining data-driven approaches from ML by using NNs with EVT is clearly inherently challenging as the extreme values are rare!
* While this work is not aligned with my area of expertise, the paper appears well-written with sufficient background to get an understanding of its significance. Code and an extensive supplementary material is provided that will help with reproducibility.
* I believe that the proposed dMNN architecture will be interesting to people in the community.

**Q4 Main Weakness:**

* The process of hyperparameter optimisation is not described and it is mentioned that the performance could be improved if they were tuned better. Are the baselines also in the same position that they could be improved with hyperparameter tuning? I realise some are parameter-free so maybe this is not an issue?
* It seems more effort was placed in showcasing dMNNs compared to the generative approach. Personally, it feels that the generative model approach is less developed from at least the empirical results. Would the paper have been better if it just focussed on the dMNNs and left the generative approach to the appendix?

**Q5 Detailed Comments To The Authors:**

* The text in the figures of the main paper are too small to read.
* For someone with less experience in this area, Figures 2 and 5 are difficult to interpret. What is the behaviour we are supposed to be looking for in these figures that makes dMNN superior (e.g. convexity etc...). It would help the reader to spell it out a bit more.
* What is a "Block" in algorithm 1.

**Q7 Justification For Your Score:**

The work is interesting and the authors seem to provide enough to be able to reproduce the results. It is also an important area to be working in. I am not confident in my score as I am outside my area of expertise.

**Q9 Complying With Reviewing Instructions:**

1: Yes.

---

### Official Review · Reviewer_jsQc · 2022-04-12

**Q2(1) Originality/Novelty:** 3
**Q2(2) Significance/Impact:** 3
**Q2(3) Correctness/Technical Quality:** 3
**Q2(6) Clarity Of Writing:** 4
**Q6 Overall Score:** 7
**Q8 Confidence In Your Score:** 2

**Q1 Summary And Contributions:**

Pickands dependence function is important to estimate extreme value copulas from data. The authors propose a fully d-max-decreasing architecture that can represent the Pickands dependence function theoretically (Theorem 2) and experimentally. Also, the authors propose a generative model that can represent the spectral measure of Pickands functions.

**Q2 Assessment Of The Paper:**

More detailed information regarding each of these aspects is given below:

**Q2(4) Quality Of Experiments (Optional):**

4: Excellent: The experimental evaluation is comprehensive and the results are compelling.

**Q2(5) Reproducibility:**

4: Excellent: Key resources (e.g., proofs, code, data) are available and key details (e.g., proof sketches, experimental setup) are comprehensively described for competent researchers to confidently and easily reproduce the main results.

**Q3 Main Strengths:**

--The paper is well written and the logic is easy to follow. There's a theory for the model and the authors also do the experiments to justify the effectiveness of their model, which seems sound to me.



**Q4 Main Weakness:**

-- I cannot find significant issues in this paper. One thing I am not very sure about is in Theorem 2: why converging to a zero-mean Gaussian process means uniform convergence. Doesn't the variance matter here?


**Q5 Detailed Comments To The Authors:**

-- In algorithm 1, is using Adam to update parameters necessary? How about SGD?

-- What's the optimization cost of the algorithm? It seems like it takes longer to train the proposed network, also there's some difficulty in tuning the hyperparameters (as mentioned by the authors).

**Q7 Justification For Your Score:**

I am not an expert in this field hence I am not very clear about the novelty of this paper. But the paper's results look solid to me. Hope other reviews can point out its significance.

**Q9 Complying With Reviewing Instructions:**

1: Yes.

---

### Official Review · Reviewer_qujv · 2022-04-16

**Q2(1) Originality/Novelty:** 3
**Q2(2) Significance/Impact:** 3
**Q2(3) Correctness/Technical Quality:** 3
**Q2(6) Clarity Of Writing:** 4
**Q6 Overall Score:** 8
**Q8 Confidence In Your Score:** 3

**Q1 Summary And Contributions:**

This manuscript proposes a neural network architecture for learning and sampling from a multivariate extreme value distribution, corresponding to the class of Pickands functions. Theoretical results show that the proposed architecture constrains the distribution to Pickands dependence functions, and that any Pickands function can be approximated by such network. After describing parameter estimation and sampling procedures, extensive validation in simulation and real data are provided.

**Q2 Assessment Of The Paper:**

More detailed information regarding each of these aspects is given below:

**Q2(4) Quality Of Experiments (Optional):**

3: Good: The experimental evaluation is adequate, and the results convincingly support the main claims.

**Q2(5) Reproducibility:**

4: Excellent: Key resources (e.g., proofs, code, data) are available and key details (e.g., proof sketches, experimental setup) are comprehensively described for competent researchers to confidently and easily reproduce the main results.

**Q3 Main Strengths:**

- the paper is very well written,
- the paper makes a comprehensive study of the question including theoretical, algorithmic and experimental aspects,
- the exhaustive documentation in the appendix is impressive,


**Q4 Main Weakness:**

- the contribution is highly technical, most likely only easily accessible to experts of extreme value distributions.

**Q5 Detailed Comments To The Authors:**

Some aspects are difficult to follow for the unfamiliar reader:
1- in practice, how to handle/choose the normalizing sequences $a_k$ and $b_k$ described in background,
2- what is the benefit of multiple layer models if one layer is enough for universal approximation? How to choose the number of layers?
3- I do not see baseline comparison, is there nothing that makes sense? I would expect that at least the low dimensional (scalar?) offers some comparison options.


**Q7 Justification For Your Score:**

A paper of high technical quality, although very specialized, with an exhaustive coverage of theoretical algorithmic and experimental aspects.

Disclaimer: I have no background knowledge of extreme value theory and as a consequence am not aware of previous work. I did go through the proofs at a superficial level and they seem technically sound.

**Q9 Complying With Reviewing Instructions:**

1: Yes.

---

### Official Review · Reviewer_YMRq · 2022-04-16

**Q2(1) Originality/Novelty:** 2
**Q2(2) Significance/Impact:** 2
**Q2(3) Correctness/Technical Quality:** 3
**Q2(6) Clarity Of Writing:** 1
**Q6 Overall Score:** 4
**Q8 Confidence In Your Score:** 1

**Q1 Summary And Contributions:**

The authors introduce a neural network architecture modelling multivariate extreme value distributions.

**Q2 Assessment Of The Paper:**

More detailed information regarding each of these aspects is given below:

**Q2(4) Quality Of Experiments (Optional):**

2: Fair: The experimental evaluation is weak: important baselines are missing, or the results do not adequately support the main claims.

**Q2(5) Reproducibility:**

1: Poor: Key details (e.g., proof sketches, experimental setup) are incomplete/unclear, or key resources (e.g., proofs, code, data) are unavailable.

**Q3 Main Strengths:**

The contribution made in the paper appears to be very specialised. As I'm not very familiar with this topic it is difficult for me to judge the merit of contributions.

**Q4 Main Weakness:**

The main weakness of this work is the lack of clarity. Most of the readers might not be familiar with MEVs and the paper is fairly poor at explaining this concept and the paper's contributions to the wider audience. Also, the paper is written in a very condensed way - I encourage the authors to provide more intuitive motivation, e.g. for the introduced architecture of the network.


**Q5 Detailed Comments To The Authors:**

To improve my understanding of the topic I've inspected the code -- it's very difficult to figure out what it is doing, and where the parts corresponding to the main contributions are implemented. I think the code clarity is especially important for topics which are not widely known, as readers can learn about the method by understanding its application to data.

The code should have its environment and clearly defined commands that can be executed to run the experiments. I also recommend the authors should use arguments to scripts as opposed to asking to "Uncomment the appropriate experiment in the bottom of the experiment".

**Q7 Justification For Your Score:**

The submission might be more suitable to a specialised journal where its technical contributions would be fully appreciated.

**Q9 Complying With Reviewing Instructions:**

1: Yes.

---

### Decision · Program_Chairs · 2022-05-15

**Decision:**

Accept (Oral)

**Comment:**

Meta Review: The paper proposes networks for high-d extreme value distributions. The reviewers were generally happy with the paper and author response. Please account for reviewer comments in your final revisions.